# New Tricks with Old Dogs: Computational Identification and Experimental Validation of New miRNA–mRNA Regulation in hiPSC-CMs

**DOI:** 10.3390/biomedicines10020391

**Published:** 2022-02-06

**Authors:** Maja Bencun, Thiago Britto-Borges, Jessica Eschenbach, Christoph Dieterich

**Affiliations:** 1Section of Bioinformatics and Systems Cardiology, Klaus Tschira Institute for Integrative Computational Cardiology, University Hospital Heidelberg, 69120 Heidelberg, Germany; Maja.Bencun@med.uni-heidelberg.de (M.B.); thiago.brittoborges@uni-heidelberg.de (T.B.-B.); Jessica.Eschenbach@med.uni-heidelberg.de (J.E.); 2DZHK (German Centre for Cardiovascular Research), Partner Site Heidelberg/Mannheim, 69120 Heidelberg, Germany; 3Department of Internal Medicine III (Cardiology, Angiology, and Pneumology), University Hospital Heidelberg, 69120 Heidelberg, Germany

**Keywords:** cardiomyocyte, stem cell, cardiovascular biology, miRNA, cardiac differentiation, beta-adrenergic stress, target prediction, miRNA–mRNA interaction

## Abstract

Cardiovascular disease is still the leading cause of morbidity and mortality worldwide. Human induced pluripotent stem cell-derived cardiomyocytes (hiPSC-CMs) have become a valuable widespread in vitro model to study cardiac disease. Herein, we employ the hiPSC-CM model to identify novel miRNA–mRNA interaction partners during cardiac differentiation and β-adrenergic stress. Whole transcriptome and small RNA sequencing data were combined to identify novel miRNA–mRNA interactions. Briefly, mRNA and miRNA expression profiles were integrated with miRNA target predictions to identify significant statistical dependencies between a miRNA and its candidate target set. We show by experimental validation that our approach discriminates true from false miRNA target predictions. Thereby, we identified several differentially expressed miRNAs and focused on the two top candidates: miR-99a-5p in the context of cardiac differentiation and miR-212-3p in the context of β-adrenergic stress. We validated some target mRNA candidates by 3′UTR luciferase assays as well as in transfection experiments in the hiPSC-CM model system. Our data show that iPSC-derived cardiomyocytes and computational modeling can be used to uncover new valid miRNA–mRNA interactions beyond current knowledge.

## 1. Introduction

Cardiovascular diseases (CVDs) are a major cause of disease burden in the world, with incidence rates on the rise [1]. Most therapy options focus on treating symptoms and improving quality of life rather than curing the disease. This is largely due to a lack of understanding of the molecular mechanisms underlying CVDs. Hypertrophic growth of cardiomyocytes is a compensatory mechanism to normalize heart performance during myocardial stress. Prolonged mechanical load on the heart can lead to progression towards pathological hypertrophy [2]. Pathological cardiac hypertrophy is a maladaptive process accompanying various forms of CVD, such as prolonged hypertension, valvular heart disease, and heart failure. Elucidating the molecular components and pathways that orchestrate the progression from pathological hypertrophy to heart failure is the focus of a lot of studies in the field of cardiology. In recent years, human induced pluripotent stem cell-derived cardiomyocytes (hiPSC-CMs) have become valuable tools to study important regulators of cardiac function and the molecular pathways underlying cardiac pathologies [3,4]. 

MicroRNAs (miRNAs) are small RNAs of approximately 22 nucleotides in length. These small endogenous RNAs regulate gene expression post-transcriptionally by targeting specific mRNAs through partial complementary base-pairing mainly in the 3′UTR of mRNAs [5]. The binding of a miRNA to its target mRNA can lead to either degradation or translational repression of the target transcript. An individual miRNA can regulate the translation of hundreds of genes and a single mRNA can, in turn, be regulated by several miRNAs making these small genetic elements important regulators of complex biological processes [6]. Two-thirds of human protein-coding genes are predicted to harbor miRNA binding sequences in their 3′UTRs [7]. They are important modulators of cardiomyocyte proliferation, differentiation, and survival [8,9,10]. Various CVDs are associated with specific miRNA expression patterns, e.g., cardiac hypertrophy, heart failure, myocardial infarction [11,12,13,14,15,16,17,18,19,20]. 

Detecting regulatory networks of functional miRNA-mRNA interactions is important to understanding the molecular basis of their action. Here, we used the in vitro model system of human induced pluripotent stem cell-derived cardiomyocytes to study miRNA expression dynamics during differentiation of hiPSCs into cardiomyocytes and during β-adrenergic receptor stimulation. The integration of transcript abundance, miRNA expression counts, and miRNA target predictions led to novel predictions of miRNA–mRNA interactions. We validated five mRNA targets of miR-99a-5p and miR-212-3p in this model system, thereby highlighting hiPSC-CMs as a useful model to study cardiac miRNA–mRNA interactions. 

## 2. Materials and Methods

### 2.1. Human Induced Pluripotent Stem Cell-Derived Cardiomyocyte (hiPSC-CM) Small RNA and Gene Expression Data

hiPSC-CM gene expression and small RNA expression data were used from a previously published study by us [21]. 

### 2.2. Bioinformatic Analyses 

We performed all bioinformatics analyses based on EnsEMBL 96 human genome and gene annotation [22]. MiRNA loci were analyzed based on miRBase v22 annotations [23].

#### 2.2.1. Differential miRNA Expression Analysis

We mapped small RNA-seq reads after adapter trimming and quality control to the human genome using bowtie2.3.5.1 (Figure 1) [24]. Prior to mapping, reads were filtered out from subsequent analysis if they contained uncalled bases or if they were shorter than 18bp after 3′ end quality clipping (Phred score < 10) and 3′ end adapter trimming. Read counts for individual miRNA loci were computed with featureCounts v1.5.1 [25]. We assessed the time dependence of read counts using the edgeR package (v3.31.4) [26]. Both experiments were analyzed independently: (1) differentiation time course: days 0, 1, 3, 5, and 15; (2) isoprenaline stimulus: 0, 4, and 24 h stimulus. All differential expression analysis results can be found in Appendix A. 

#### 2.2.2. miRNA–mRNA Prediction Analysis

We computed transcript abundance estimates (FPKM) with stringtie 1.3.5 [29] and merged all time points for the 2 independent analyses using ballgown 2.22 (Figure 1) [30].

We retrieved conserved miRNA target site predictions from TargetScan7.1 (http://www.targetscan.org/vert_70/ (accessed on 7 January 2018)) [31] and integrated them with normalized miRNA read counts and transcript abundance estimates. The general bioinformatics concept based on the global test statistics is detailed in van Iterson et al. [28] and was performed here in the same way except for using different input data as outlined above. Global test results are summarized in Appendix A. 

### 2.3. HiPSC Differentiation into Cardiomyocytes

The WT1.14 cells used for differentiation into cardiomyocytes were a kind gift from Shirin Doroudgar. All cells were maintained in a 37 °C humidified incubator with 5% CO_2_. Human induced pluripotent stem cells were grown on Matrigel (BD Bioscience, San Jose, CA, USA)-coated 6-well plates to 90–95% confluency in Stem-MACS iPS Brew XF Medium (Miltenyi Biotec, Bergisch Gladbach, Germany) supplemented with Stem MACS iPS Brew XF Supplement (Miltenyi Biotec, Bergisch Gladbach, Germany). At day 0 (90–95% confluency of cells) the medium was switched to RPMI 1640 with HEPES and GlutaMax (Thermo Fisher Scientific, Waltham, MA, USA), B27 Supplement minus Insulin (Thermo Fisher Scientific, Waltham, MA, USA) and 4 µM CHIR99021 (Merck Millipore, Burlington, MA, USA). Twenty-four hours later, 3 mL of fresh medium were added to the cells. Three days post differentiation start, the medium was replaced with RPMI 1640 with HEPES and GlutaMax supplemented with B27 Supplement minus Insulin and 2.5 µM IWP2 (Merck Millipore, Burlington, MA, USA). On day 5, medium was changed to RPMI 1640 with HEPES and GlutaMax plus B27 Supplement minus Insulin. On day 7, the media was switched to RPMI 1640 with HEPES and GlutaMax plus B27 supplement and was changed every 2 days until day 10 when spontaneous beating could be observed throughout the culture. The cells were purified by a lactate metabolic selection method. Briefly, the selection medium was prepared with RPMI 1640 without Glucose (Thermo Fisher Scientific, Waltham, MA, USA) supplemented with human recombinant Albumin (Sigma-Aldrich Chemie GmbH, Taufkirchen, Germany), L-Ascorbic Acid 2-Phosphate (Sigma-Aldrich Chemie GmbH, Taufkirchen, Germany), and 4 mM Sodium DL-lactate solution (Sigma-Aldrich Chemie GmbH, Taufkirchen, Germany). Cells were cultured in this selection medium for four days before switching back to cardio culture medium (RPMI 1640 with Glucose, GlutaMax, HEPES, and B27 supplement). Lactate metabolic selection was performed twice before plating the cells for experiments. 

### 2.4. RNA Extraction 

Cells were lysed in Quiazol reagent (Qiagen GmbH, Hilden, Germany) according to manufacturer’s instructions and RNA was extracted using the miRNeasy Mini kit (Qiagen GmbH, Hilden, Germany). For the hiPSC differentiation, time-course RNA samples were taken at days 0, 5, 7, 9, 11, and 20 in three independent differentiation experiments. 

### 2.5. Real Time Quantitative PCR (RT-qPCR) 

A total of 500 ng of total RNA was reverse transcribed using iScript cDNA synthesis kit (Bio-Rad Laboratories GmbH, Feldkirchen, Germany) according to manufacturer’s instructions. Then, 7.5 ng of cDNA was used for RT-qPCR analysis with iTaq Universal SYBR Green Supermix (Bio-Rad Laboratories GmbH, Feldkirchen, Germany). Primers used for reverse transcription and RT-qPCR are listed in Appendix A. 

### 2.6. Real Time Quantitative PCR for miRNA Expression Levels 

The RevertAid First Strand cDNA synthesis kit (Thermo Fisher Scientific, Waltham, MA, USA) was used to reverse transcribe 250 ng total RNA. The stem-loop reverse transcription protocol described by Wang et al. was used to reverse transcribe miRNAs [32]. In total, 9 ng of cDNA was used for RT-qPCR analysis with iTaq Universal SYBR Green Supermix (Bio-Rad Laboratories GmbH, Feldkirchen, Germany) for specific miRNA detection. The relative miRNA expression levels were calculated using the comparative Ct method 2^−ΔΔCt^ [33]. The small RNA U6 was used for normalization. Primers used for reverse transcription and RT-qPCR are listed in Appendix A. 

### 2.7. miRNA Mimic and Inhibitor Transfections

HiPSC-CMs were maintained in cardio culture medium (RPMI 1640 with Glucose, GlutaMax, HEPES, and B27 supplement) and incubated at 37 °C in a humidified incubator with 5% CO_2_. For transient transfection experiments hiPSC-CMs were washed briefly with Versene solution (Life Technologies, Thermo Fisher Scientific, Waltham, MA, USA), detached using 0.25% Trypsin-EDTA solution, washed once in splitting medium (RPMI 1640 with HEPES and GlutaMax, B27 supplement, 20% FCS (Thermo Fisher Scientific, Waltham, MA, USA), 2 µM Thiazovivin (Merck Millipore, Burlington, MA, USA)) and seeded at the desired cell density. Twenty-four hours after seeding, the cells were switched back to cardio culture medium, allowed to recover for 48 h, and then used for experiments. To modify the expression levels of specific miRNAs, the cells were transfected with miRNA mimics or inhibitors (Qiagen GmbH, Hilden, Germany) at 20 and 50 nM concentrations, respectively. Lipofectamine RNAiMax (Thermo Fisher Scientific, Waltham, MA, USA) was used as the transfection reagent according to the manufacturer’s instructions. All miRNA mimics and inhibitors as well as the corresponding negative controls were purchased from Qiagen (Qiagen GmbH, Hilden, Germany). An overview of miRNA mimics and inhibitors used in this study is listed in Appendix A. 

### 2.8. Dual-Luciferase Reporter Assays

HEK293 cells were used for transient transfection of luciferase 3′UTR reporter constructs. The cells were cultured in DMEM (Life Technologies, Thermo Fisher Scientific, Waltham, MA, USA) supplemented with 10% FCS (Gibco, Waltham, MA, USA). For transient transfection experiments using luciferase, constructs cells were seeded on 12-well plates. At 70% confluency, luciferase reporter plasmids were transfected using Viafect (Promega, Madison, WI, USA) according to the manufacturer’s instructions. In parallel, miRNA mimics or scrambled controls were transfected at a final concentration of 50 nM using Lipofectamine RNAiMax (Thermo Fisher Scientific, Waltham, MA, USA) as described in Section 2.7. Cells were lysed 24 h after transfection and luciferase measurements were performed using the Dual Luciferase Reporter Assay (Promega, Madison, WI, USA) according to the manufacturer’s instructions. *Firefly* luciferase expression was normalized to *Renilla* luciferase activity. The 3′UTR reporter plasmids were purchased from BioCat GmbH, Heidelberg, Germany. An overview of reporter plasmids used in this study is listed in Appendix A. 

### 2.9. HiPSC-CM β-Adrenergic Stimulation

Cells were detached and seeded on 12-well plates as described above (Section 2.8 miRNA mimic and inhibitor transfections). The cells were allowed to recover for at least 48 h post seeding and were stimulated with 5 µM isoprenaline (Sigma-Aldrich Chemie GmbH, Taufkirchen, Germany) for 24 h. 

## 3. Results

### 3.1. MicroRNA Expression Profiling during Human Induced Pluripotent Stem Cell (hiPSC) Differentiation into Cardiomyocytes

In a previous study, we characterized the gene expression changes observed during cardiomyocyte differentiation from hiPSCs and confirmed their suitability for the analysis of molecular events regarding cardiac lineage development, fetal gene program activation, as well as cardiac response to stimuli [21]. Here, we compared the whole transcriptome with small RNA sequencing of hiPSCs undergoing differentiation (Figure 2A). We identified important expression changes in miRNA and mRNA expression levels at different stages of cardiac lineage development. We computed a global test that identifies sets of significantly correlated target genes from these candidates, allowing us to find novel miRNA–mRNA interaction networks. We ended up with 199 different miRNAs regulated during the experiment (Appendix A). The number of predicted mRNA targets for individual miRNAs varied between 1 and 778 genes (see Appendix A).

Our data set readily identified the upregulation of key miRNAs known to be associated with cardiac muscle differentiation, e.g., miR-133b [8,9,34,35] (Table 1 and Appendix A). Furthermore, miRNAs reported to regulate contractile protein expression such as miR-208a-3p, miR-499a-5p, as well as miRNAs of the miR-15 family that control cell cycle genes, were upregulated during hiPSC differentiation into cardiomyocytes (Appendix A) [8,9,35]. 

The global test on differential expression analysis of miRNAs identified miR-99a-5p as one of the top 20 most significant miRNAs regulated during differentiation (Table 1) [36]. We collected RNA samples on different days after the start of hiPSC differentiation and assessed miRNA and mRNA expression levels by real-time qPCR (RT-qPCR). We experimentally confirmed an increase in expression levels of miR-99a-5p as the hiPSCs transition from cardiac mesoderm to cardiac progenitors and finally to cardiomyocytes (Figure 2A–C and Appendix A). 

The network table of predicted miR-99a-5p targets includes 5 mRNAs (Table 2 and Appendix A). The mRNAs with the strongest negative association with miR-99a-5p expression include RNA-binding proteins, solute carriers, and transcriptional regulators. We chose two of these target mRNAs to validate further: tripartite motif containing 71 (TRIM71) and solute carrier 44A1 (SLC44A1). These two target mRNAs showed a strong negative correlation with miR-99a-5p expression levels (Figure 2B). Analysis of transcript abundance during hiPSC differentiation into cardiomyocytes confirmed our sequencing data (Figure 2D). Both predicted mRNA targets, TRIM71 and SLC44A1, decreased in expression levels significantly as the miRNA level increased (Figure 2C,D and Appendix A). The known miR-99a-5p target mTOR also decreased during the differentiation of hiPSCs to hiPSC-CMs but not as significantly as our two predicted targets (Figure 2D) [37,38]. 

### 3.2. MicroRNA Expression Profiling after β-Adrenergic Stimulation

We also computed a global test for late hiPSC-CMs undergoing chronic β-adrenergic signaling. Chronic β-adrenergic receptor stimulation was induced by exposing the cells to 5 µM isoprenaline for 24 h. We ended up with 168 different miRNAs regulated during the experiment (Appendix A). MiRNAs known to be upregulated during β-adrenergic signaling in cardiac cells are miR-132-3p and miR-212-5p [13]. In our experiment, the global test on the isoprenaline stimulus identified miR-212-3p as differentially expressed following chronic stress (Table 3). On the other hand, miR-132-3p expression level changes were not ranked as significant (Appendix A) [13]. In general, miR-132-3p expression levels were low in our hiPSC-CMs (Appendix A). We did find other miRNAs known to be highly expressed in the heart during cardiac hypertrophy, e.g., miR-208a-3p and miR-21-5p, miR-23a, miR-23b [12,39] (Table 3 and Appendix A). Finally, we decided to study miR-212-3p and its predicted targets closer. Only two miR-212-3p targets are predicted as significant in our database: Talin 2 (TLN2) and DAZ associated protein 2 (DAZAP2) (Table 4, Figure 3A). Although not ranked as significant in the global test, we decided to include transmembrane protein 2 (TMEM2) in our validation studies of miR-212-3p mRNA targets (Figure 3A). We exposed hiPSC-CMs to 5 µM isoprenaline for 24 h again and measured the expression levels of our target transcripts by RT-qPCR. Upon isoprenaline stimulation the cardiac hypertrophy markers atrial natriuretic peptide (ANP) and brain natriuretic peptide (BNP) were significantly increased in our cells (Figure 3B,C). The predicted miR-212-3p mRNA targets TLN2 and DAZAP2 were decreased in their relative expression level upon stimulation (Figure 3D). We did not see a decrease in TMEM2 mRNA levels (Figure 3D). Two experimentally studied miR-212-3p targets AGO2 [40] and ZEB2 [41] did not decrease in their expression level either (Figure 3D). This further highlights the cell type dependent function of miRNAs. 

### 3.3. Validation of miR-99a-5p and miR-212-3p Target Genes in Reporter Assays

Next, we sought to show that our chosen miRNAs miR-99a-5p and miR-212-3p directly regulated our predicted target mRNAs. The 3′ untranslated regions (UTRs) of our chosen mRNA targets were cloned after the coding region of the *Firefly* luciferase gene. *Renilla* luciferase encoded on the same plasmid was used for transfection normalization. These expression constructs were co-transfected with either the miRNA mimic (miR-99a-5p, miR-212-3p) or a scrambled miRNA control (miR-Scr) into HEK293 cells. For the two targets of miR-99a-5p we could confirm a significant downregulation of *Firefly* luciferase activity upon miRNA mimic transfection (Figure 4A). The predicted targets DAZAP2 and TMEM2 were also significantly downregulated upon miR-212-3p mimic transfection (Figure 4B). We did not observe a downregulation of the Luc-TLN2 reporter construct in our transfection experiments (Figure 4B). 

### 3.4. Validation of miR-99a-5p and miR-212-3p Target Genes in hiPSC-CMs

A single miRNA can regulate multiple even hundreds of mRNA transcripts simultaneously. The extent of the regulation is cell type dependent. Therefore, we wanted to confirm that the mRNA targets predicted for miR-99a-5p and miR-212-3p were indeed regulated in hiPSC-CMs. 

We transfected synthetic miR-99a-5p mimic into hiPSC-CMs at 14 and 34 days post differentiation start (Figure 5A,B). At 14 days post differentiation start we could confirm the downregulation of the newly identified mRNA target transcripts (TRIM71, SLC44A1) as well as known miR-99a-5p targets (AGO2) (Figure 5A,B). When we transfected the synthetic miRNA mimic into older hiPSC-CMs we did not see a downregulation of TRIM71 or AGO2. Transfection of miR-99a-5p inhibitor at 34 days post differentiation start on the other hand led to a significant upregulation of these targets in hiPSC-CMs (Figure 5C). 

We also transfected the miR-212-3p mimic into hiPSC-derived cardiomyocytes. This led to a downregulation of known (AGO2) as well as of newly identified targets (TMEM2, DAZAP2, TLN2) (Figure 5D). Transfection of the miR-212-3p inhibitor in turn led to an increase in target transcript levels (Figure 5E). 

## 4. Discussion

Our current study expands our previously published characterization of the hiPSC-CM model system by looking at miRNA and mRNA expression dynamics [21]. 

We provide here an overview and catalogue of dynamically regulated miRNAs and their negatively associated mRNA targets during cardiac differentiation and β-adrenergic stimulation. Our experiments show that exogenous transfection of our studied miRNAs leads to a decrease in the predicted target mRNA levels in hiPSC-CMs as well as in luciferase assays.

The TRIM71 protein promotes stem cell proliferation and is strongly downregulated during exit from pluripotency [42,43,44]. We measured TRIM71 levels following miR-99a-5p mimic transfections at an early and late timepoint post hiPSC differentiation start. The early timepoint showed a significant downregulation of TRIM71 mRNAs after mimic transfection while the later timepoint did not. This is most likely due to already very low TRIM71 mRNA levels in fully differentiated hiPSC-CMs. 

TLN2, one of the predicted targets of miR-212-3p, showed a significant downregulation in expression levels upon β-adrenergic stimulation but did not show a downregulation in the luciferase reporter assays. Nevertheless, we observed a mild downregulation of TLN2 following miR-212-3p mimic transfections in hiPSC-CMs. The luciferase assays were performed in HEK293 cells, a cell line of epithelial morphology further highlighting the highly cell type specific function of miRNAs. 

This was also observed with another miR-212-3p target: TMEM2. On the one hand, we did not observe a reduction in TMEM2 mRNA levels during isoprenaline stimulation but could see a significant downregulation in the luciferase reporter assays. This confirms that miR-212-3p does target the 3′UTR of TMEM2 and offers a partial explanation why this mRNA was not identified as a highly significant target in our hiPSC-CM sequencing data. Furthermore, it confirms the validity of our target predictions. 

Just as understanding the gene expression changes that occur during disease development is important, it is also necessary to understand the molecular basis of cardiomyocyte development as this can inform on the mechanisms of cardiac pathogenesis. Interestingly, the miRNA-99/-100 family is downregulated in zebrafish following cardiac injury [45]. Cardiac regeneration can artificially be triggered in mice by reducing miR-99 and miR-100 levels following myocardial infarction [45]. 

Identification of disease-related miRNAs has increasingly become an important goal in the biomedical research on CVDs. It not only accelerates the understanding of underlying disease pathogenesis at the molecular level but also offers novel therapeutic approaches to treat and diagnose as well as potentially prevent the disease itself. 

The predicted miRNA–mRNA pairs identified in our study offer biological guidance for investigating this gene expression regulatory system in the context of cardiac development and disease associations. The experimental system utilizing isoprenaline stimulation readily identified miRNAs reported to be upregulated in HCM and heart failure patients as well as in studies of cardiac hypertrophic growth (Figure 6) [46,47]. Even though the importance of altered miRNA levels and the resulting deregulation of gene expression in the development of CVD pathology is known, there are still many gaps in our knowledge. This is due to the large number of mRNAs that can be regulated by a single miRNA. Identifying and understanding the relevant mRNAs in a specific cellular context is still a matter of ongoing research. Furthermore, the combination of several miRNAs targeting the same mRNA will also influence gene expression and the downstream associated pathways, further complicating discovery of relevant targets.

The role of miRNAs in human CVDs not only aids in the understanding of the underlying molecular mechanisms of disease development but offers up novel tools for biomarker identification. Cells secrete miRNAs through exosomes and microvesicles into the circulation [48,49]. These miRNAs are surprisingly stable and can readily be detected in blood samples. Circulating miRNAs are currently studied as potential biomarkers in various CVDs by characterizing specific miRNA blood profiles [50]. Particularly, miR-21 has consistently been reported as circulating in the blood of HCM patients at elevated concentrations, making it an interesting biomarker target (Figure 6) (reviewed in [46]). Nevertheless, the biomarker aspect of miRNAs is still in the early development stage. It is still unclear what the cutoff values for miRNA concentrations are in patients vs. controls. Studies have been heterogeneous, and so far, only two miRNAs have consistently been identified in blood samples of HCM and heart failure patients compared to controls: miR-21 and miR-29 (Figure 6) [46]. Larger patient cohort studies are necessary to fully develop the diagnostic potential of circulating miRNAs. Our miRNA-target gene studies can be altered further to include miRNA–disease associations which can then be used to identify miRNAs that could function as biomarkers in patients [51,52]. 

As miRNAs function in a sequence-specific manner to exert gene expression control, they are good drug targets and are being explored as novel therapeutics to treat CVDs. Therapies based on modulation of miRNA levels in target tissues aim at increasing miRNA levels through delivery of a miRNA mimic or siRNA. On the other hand, miRNA levels may need to be downregulated. This is achieved through the delivery of synthetic miRNA antagonists termed antagomirs. Antagomirs are chemically modified oligonucleotides that bind to the mature miRNA and thereby block its binding to the target mRNA [53]. 

Apart from being readily detectable in HCM patient blood samples [54,55], miR-21 is also consistently upregulated in cardiac tissue in these patients [18,56]. This finding makes miR-21 an interesting therapeutic target to treat heart failure. Unfortunately, miR-21 has proven unsuitable as a therapeutic target as its inhibition in mice did not have a beneficial effect on hypertrophic growth following thoracic aortic constriction [57]. Other miRNAs are also considered as therapeutic targets for the treatment of heart failure (Figure 6) [58]. The miR-212/132 family is an important regulator of cardiac hypertrophy and heart failure development in experimental model systems as well as in vivo [13]. The upregulation of these miRNAs is sufficient to direct hypertrophic growth in cardiomyocytes. An antisense oligonucleotide targeting miR-132, CDR132L, has entered early clinical trials in humans for the treatment of heart failure [59]. Another miRNA-based antisense oligonucleotide has also shown promising results in wound healing by targeting miR-92a and is considered for use following myocardial infarction [60].

One aspect we have not analyzed in our study is the observation that some miRNAs could also function as positive regulators of gene expression control [61,62]. We expect that our data presents itself as an informative resource for discovering miRNA dynamics during the pathogenesis of stress and cardiac lineage development. 

## Figures and Tables

**Figure 1 biomedicines-10-00391-f001:**
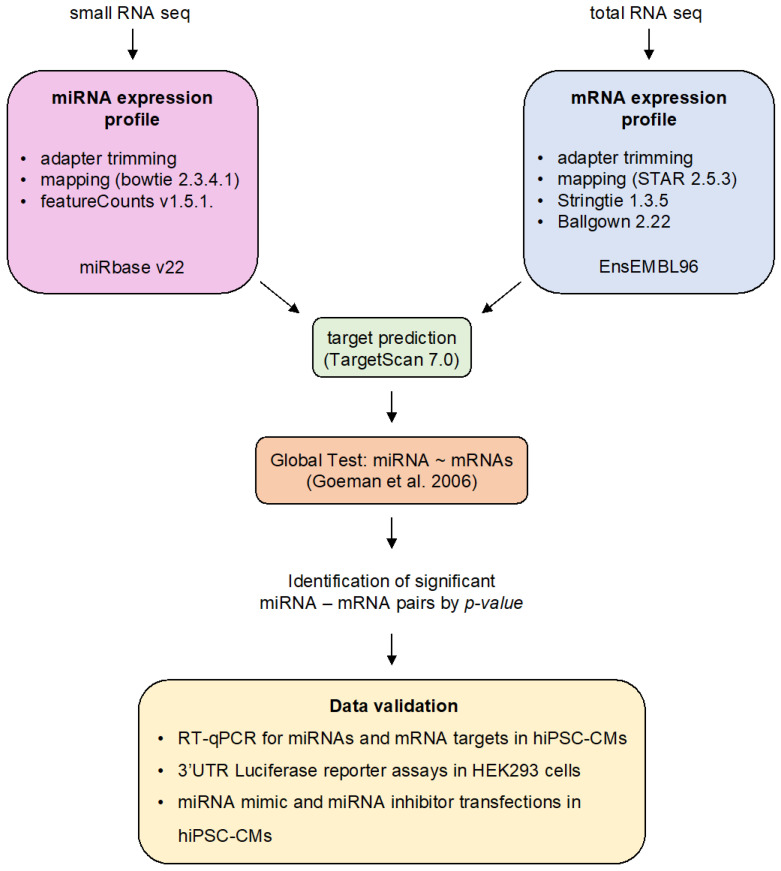
Overview of the bioinformatic analysis workflow and experimental validation methods used in this study. The small RNA and total RNA sequencing data were obtained from a previously published study [21]. Expression profiles of miRNAs and mRNAs were computed as outlined in the figure. We performed an integrated analysis of miRNA and mRNA expression data based on the global test developed by Goeman et al. [27] and as detailed by van Iterson et al. for miRNA target prediction [28]. Data validation was performed in vitro in HeLa cells or human induced pluripotent stem cell-derived cardiomyocytes (hiPSC-CMs).

**Figure 2 biomedicines-10-00391-f002:**
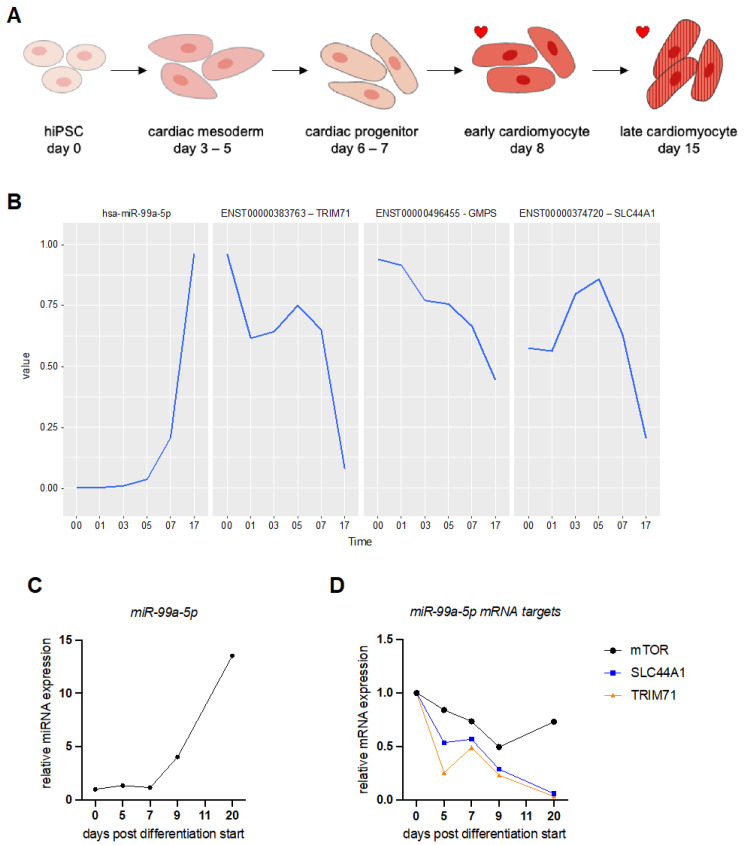
Expression level of miR-99a-5p and its putative target mRNAs during human induced pluripotent stem cell differentiation into cardiomyocytes. (**A**) Schematic overview of cardiomyocyte differentiation from human induced pluripotent stem cells (hiPSC). Stages where beating cells can be observed are marked by a red heart. (**B**) The diagram shows the changing expression levels of miR-99a-5p and three of its predicted mRNA target transcripts extracted from the sequencing data. The expression dynamics were profiled starting from day 0 and end at day 17 (Time axis). Depicted are from left to right: hsa-miR-99a-5p expression levels (panel 1), ENST00000383763 (TRIM71) mRNA levels (panel 2), ENST00000496455 (GMPS) mRNA levels (panel 3), and ENST00000374720 (SLC44A1) mRNA levels (panel 4). (**C**) The diagram shows the changing expression levels of miR-99a-5p during hiPSC differentiation into cardiomyocytes as measured by RT-qPCR. One representative experiment is shown. For further replicates see Appendix A. (**D**) The diagram shows the changing expression levels of miR-99a-5p mRNA target transcripts during hiPSC differentiation into cardiomyocytes as measured by RT-qPCR. One representative experiment is shown. For additional replicates, see Appendix A.

**Figure 3 biomedicines-10-00391-f003:**
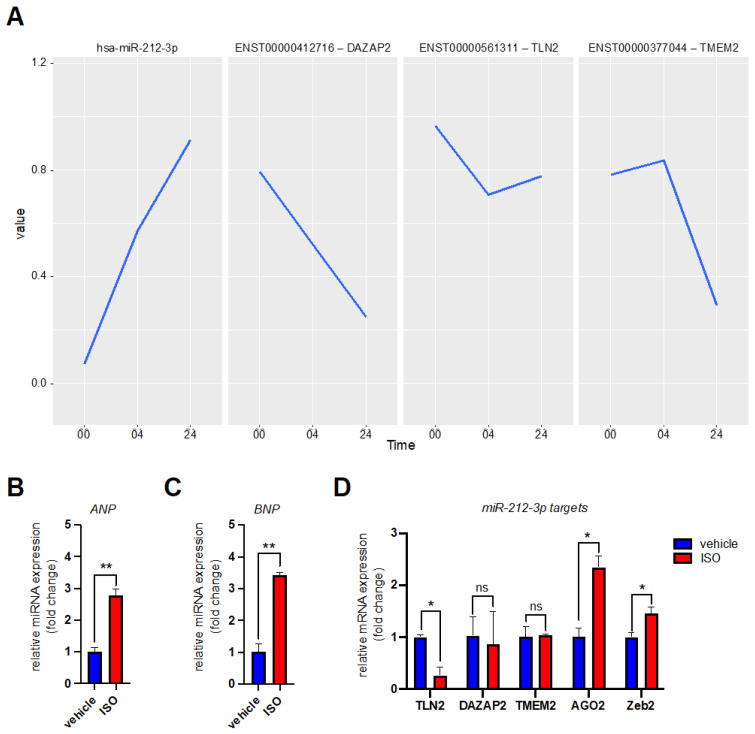
Expression level of miR-212-3p and its putative target mRNAs during β-adrenergic stimulation of hiPSC-CMs. (**A**) The diagram shows the changing expression levels of miR-212-3p and three of its predicted mRNA target transcripts extracted from the sequencing data. The *x*-axis (Time) represents the duration of β-adrenergic receptor stimulation exerted on the cells (in hours). Depicted are from left to right: hsa-miR-212-3p expression levels (panel 1), and its predicted mRNA target transcripts ENST00000412716 (DAZAP2) (panel 2), ENST00000561311 (TLN2) (panel 3), and ENST00000377044 (TMEM2) (panel 4). (**B**) The bar graph shows atrial natriuretic peptide (ANP) mRNA levels in hiPSC-CMs under physiologic conditions (vehicle) and during β-adrenergic receptor stimulation (isoprenaline: ISO). Isoprenaline-mediated adrenergic stress was applied for 24 h. The bar graphs represent two independent biological replicates. (**C**) The bar graph shows brain natriuretic peptide (BNP) mRNA levels in hiPSC-derived cardiomyocytes under physiologic conditions (vehicle) and during β-adrenergic receptor stimulation (isoprenaline: ISO). Isoprenaline-mediated adrenergic stress was applied for 24 h. The bar graphs represent two independent biological replicates. (**D**) The bar graph shows the mRNA levels of miR-212-3p targets in hiPSC-CMs under physiologic conditions (vehicle) and during β-adrenergic receptor stimulation (isoprenaline: ISO). Isoprenaline-mediated adrenergic stress was applied for 24 h. The bar graphs represent two independent biological replicates. Data was analyzed by unpaired *t*-test. Error bars indicate ±standard deviation; ns: not significant; * *p*-value ≤ 0.05; ** *p*-value ≤ 0.01.

**Figure 4 biomedicines-10-00391-f004:**
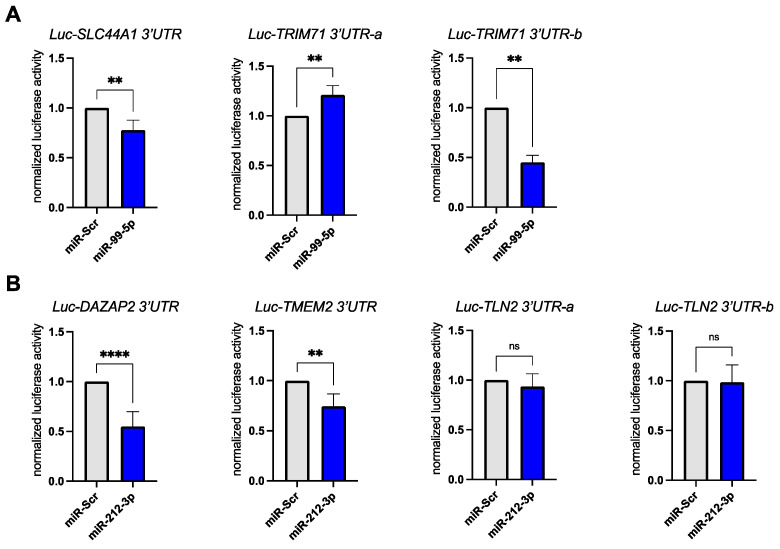
Luciferase 3′UTR reporter assays confirm miRNA–mRNA target pairs. (**A**) The bar graph shows luciferase activity levels in HEK293 cells following co-transfection of luciferase 3′UTR reporter plasmids with miRNA mimics (miR-Scr as scrambled control and synthetic miR-99a-5p). Each experiment was performed with a minimum of 4 experiments per group. *Firefly* luciferase activity was normalized to the internal control: *Renilla* luciferase. Note: In cases where the 3′UTR of the mRNA target was longer than 3.5 kb, the UTR was broken up into shorter fragments and cloned into two constructs with overlapping sequence (e.g., TRIM71). (**B**) The experiment was performed as in A using synthetic miR-212-3p and its predicted mRNA targets. Each experiment was performed with a minimum of 4 experiments per group. *Firefly* luciferase activity was normalized to the internal control: *Renilla* luciferase. Note: In cases where the 3′UTR of the mRNA target was longer than 3.5 kb, the UTR was broken up into shorter fragments and cloned into two constructs with overlapping sequence (e.g., TLN2). Data was analyzed by unpaired *t*-test. Error bars indicate ±standard deviation; ns: not significant; ** *p*-value ≤ 0.01; **** *p*-value ≤ 0.0001.

**Figure 5 biomedicines-10-00391-f005:**
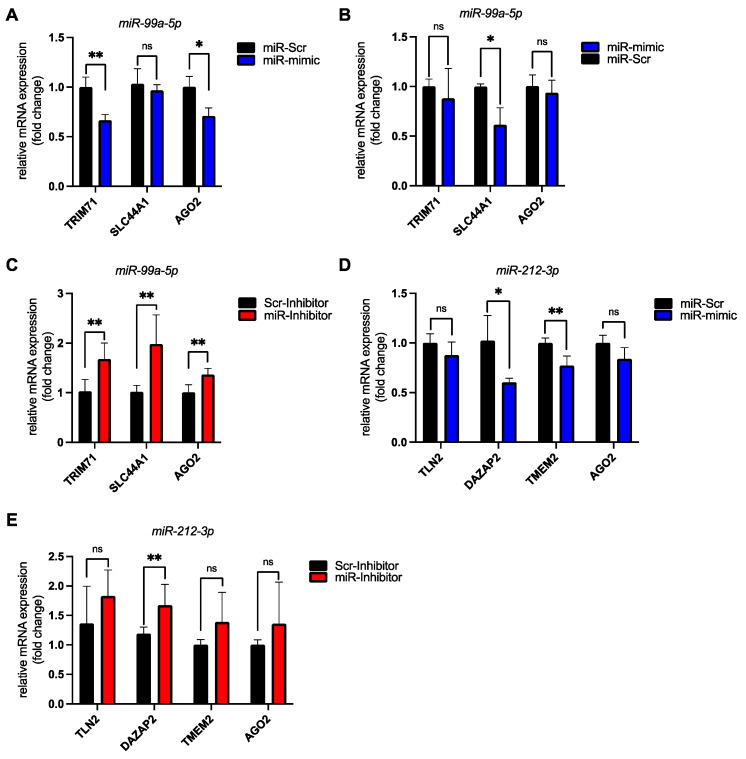
Validation of miRNA–mRNA target pairs in human induced pluripotent stem cell-derived cardiomyocytes (hiPSC-CMs) by miRNA mimic and inhibitor transfections confirms our in silico predictions. (**A**) HiPSC-CMs were transfected 14 days post differentiation start with 20nM of either a scrambled miRNA control (miR-Scr) or synthetic miR-99a-5p (miR-mimic). The bar graph shows relative mRNA expression levels of predicted miR-99a-5p targets (TRIM71, SLC44A1) and a confirmed target (AGO2) as assessed by RT-qPCR. n = 3 biological replicates. (**B**) HiPSC-CMs were transfected 34 days post differentiation start with 20nM of either a scrambled miRNA control (miR-Scr) or synthetic miR-99a-5p (miR-mimic). The bar graph shows relative mRNA expression levels of predicted miR-99a-5p targets (TRIM71, SLC44A1) and a confirmed target (AGO2) as assessed by RT-qPCR. n = 3 biological replicates. (**C**) HiPSC-CMs were transfected 34 days post differentiation start with 50 nM of either a control miRNA inhibitor (Scr-Inhibitor) or a specific miR-99a-5p inhibitor (miR-Inhibitor). The bar graph shows relative mRNA expression levels of predicted miR-99a-5p targets (TRIM71, SLC44A1) and a confirmed target (AGO2) as assessed by RT-qPCR. n = 6 biological replicates. (**D**) HiPSC-CMs were transfected 34 days post differentiation start with 20 nM of either a scrambled miRNA control (miR-Scr) or synthetic miR-212-3p (miR-mimic). The bar graph shows relative mRNA expression levels of predicted miR-212-3p targets (TLN2, DAZAP2, TMEM2) and a confirmed target (AGO2) as assessed by RT-qPCR. n = 4 biological replicates. (**E**) HiPSC-CMs were transfected 34 days post differentiation start with 50 nM of either a control miRNA inhibitor (Scr-Inhibitor) or a specific miR-212-3p inhibitor (miR-Inhibitor). The bar graph shows relative mRNA expression levels of predicted miR-212-3p targets (TLN2, DAZAP2, TMEM2) and a confirmed target (AGO2) as assessed by RT-qPCR. n = 8 biological replicates. Data was analyzed by unpaired *t*-test. Error bars indicate ±standard deviation; ns: not significant; * *p*-value ≤ 0.05; ** *p*-value ≤ 0.01.

**Figure 6 biomedicines-10-00391-f006:**
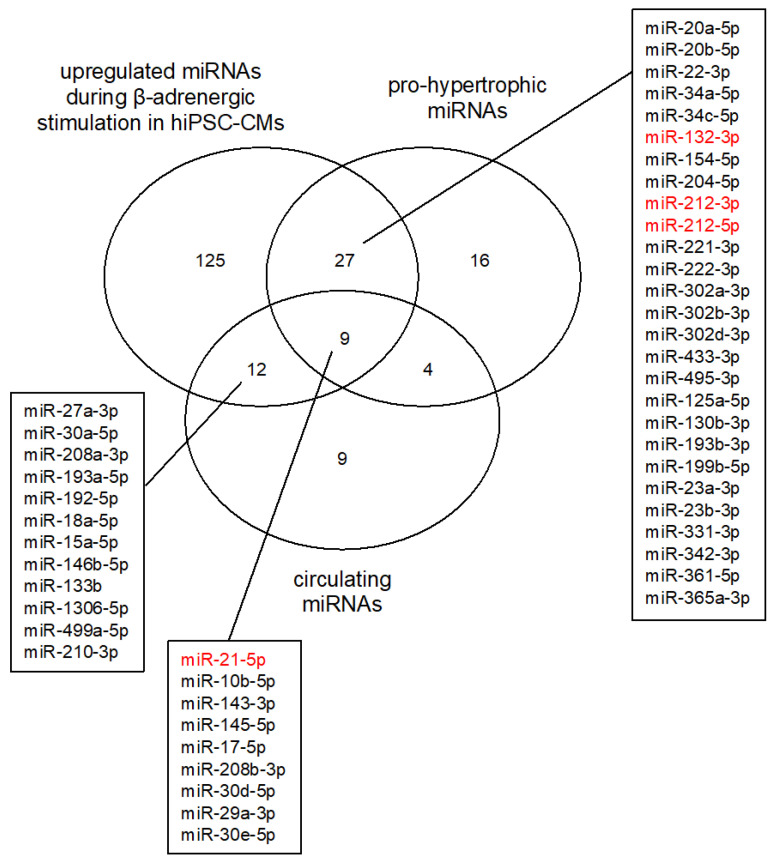
Comparison of miRNA expression in hiPSC-CMs undergoing β-adrenergic stimulation to circulating and cardiac miRNAs identified in HCM and heart failure patients. The Venn diagram shows the number of miRNAs upregulated in (i) hiPSC-CMs undergoing β-adrenergic stimulation, (ii) circulating miRNAs identified in HCM and heart failure patients (circulating miRNAs), as well as (iii) cardiac miRNAs upregulated in HCM and heart failure patients or under hypertrophic conditions (pro-hypertrophic mRNAs). Upregulated miRNAs identified in this study that have been observed in either or both of the other categories are listed. Highlighted in red are the miRNAs that are studied as therapeutic targets. The studies used to generate the figure were taken from the following three reviews: [11,46,47]. Only miRNAs studied in the context of HCM, heart failure, and hypertrophic growth were included here.

**Table 1 biomedicines-10-00391-t001:** List of miRNAs showing the strongest association with predicted mRNA targets in the global test during the differentiation time course.

miRNA	*p*-Value	Number of mRNA Targets
hsa-let-7c-5p	4.97331 × 10^−12^	6
hsa-miR-31-5p	3.70845 × 10^−10^	221
hsa-miR-489-3p	7.58567 × 10^−10^	162
hsa-miR-490-3p	1.08922 × 10^−7^	129
hsa-miR-27b-3p	2.98507 × 10^−7^	64
hsa-miR-181d-5p	3.3065 × 10^−7^	156
hsa-miR-362-5p	4.79262 × 10^−7^	72
hsa-miR-28-3p	5.71315 × 10^−7^	74
hsa-miR-34c-5p	7.34698 × 10^−7^	132
hsa-miR-23b-3p	2.08809 × 10^−6^	37
hsa-miR-146a-5p	2.1465 × 10^−6^	55
hsa-miR-99b-5p	2.89621 × 10^−6^	5
hsa-miR-142-5p	3.16525 × 10^−6^	238
hsa-miR-302d-3p	4.01342 × 10^−6^	2
hsa-miR-99a-5p	5.67015 × 10^−6^	17
hsa-miR-449a	8.3154 × 10^−6^	26
hsa-miR-30a-5p	8.33399 × 10^−6^	197
hsa-miR-181c-5p	1.49425 × 10^−5^	88
hsa-miR-335-5p	2.00942 × 10^−5^	156
hsa-miR-30b-5p	2.33941 × 10^−5^	317

**Table 2 biomedicines-10-00391-t002:** Strongest negatively expressed mRNA targets predicted to be regulated by miR-99a-5p during induced pluripotent stem cell differentiation into hiPSC-CMs.

mRNA Target	*p*-Value	z-Score
TRIM71	1.86 × 10^−6^	8.7854311
GMPS	1.432 × 10^−5^	7.967437
SLC44A1	0.0002668	6.3925893
RAVER2	0.0172793	3.0612766
ETV3	0.04815	2.0275677

**Table 3 biomedicines-10-00391-t003:** List of miRNAs showing the strongest association with predicted mRNA targets in the global test during β-adrenergic stimulation.

miRNA	*p*-Value	Number of mRNA Targets
hsa-miR-191-5p	0.0013	34
hsa-miR-503-5p	0.0014	219
hsa-miR-140-5p	0.0022	238
hsa-miR-370-5p	0.0033	143
hsa-miR-296-3p	0.0043	41
hsa-miR-208b-3p	0.0056	67
hsa-miR-212-3p	0.0076	92
hsa-miR-539-3p	0.0106	223
hsa-miR-30b-5p	0.0124	302
hsa-let-7d-5p	0.0147	198
hsa-miR-365a-3p	0.0155	179
hsa-miR-208a-3p	0.0159	59
hsa-miR-1271-5p	0.0165	367
hsa-miR-34a-5p	0.0167	87
hsa-miR-21-5p	0.0173	194
hsa-miR-99b-5p	0.019	6
hsa-miR-145-5p	0.0221	10
hsa-miR-125a-5p	0.0258	166
hsa-miR-20a-5p	0.0271	192
hsa-miR-17-5p	0.0331	128

**Table 4 biomedicines-10-00391-t004:** The two predicted mRNA targets regulated by miR-212-3p in hiPSC-CMs.

mRNA Target	*p*-Value	z-Score
DAZAP2	0.0021	3.8486
TLN2	0.0338	2.253

## Data Availability

All sequencing data are available under NCBI Bioproject ID: https://www.ncbi.nlm.nih.gov/bioproject/PRJNA391928 (accessed on 26 June 2017).

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
