# Peer review of "New Tricks with Old Dogs: Computational Identification and Experimental Validation of New miRNA–mRNA Regulation in hiPSC-CMs"

_biomedicines, 2022, doi:10.3390/biomedicines10020391_

Round 1

Reviewer 1 Report

I revised the manuscript entitled “New Tricks with Old Dogs: Computational Identification and Experimental Validation of New miRNA-mRNA Regulation in hiPS-CMs”.

This is a well-written and well-balanced essay.

I have only some minor points to comment:

  • Some typographical errors need to be corrected (for example: section 1, last paragraph “into cardiomyocytes und during β-adrenergic”, or the different font in the figure 3 legend)
  • Table 3 and 2 appear into the text before table 1
  • It would be of value to include a figure summarizing and depicting the role of miRNAs in CVD pathophysiology along with their potential as biomarkers or therapeutic targets.

That’s all I have only to comment for this elegant piece of work.

Author Response

Reviewer #1

We are grateful to reviewer #1 for the positive and constructive feedback we have received

I revised the manuscript entitled “New Tricks with Old Dogs: Computational Identification and Experimental Validation of New miRNA-mRNA Regulation in hiPS-CMs”.

This is a well-written and well-balanced essay.

I have only some minor points to comment:

  1. Some typographical errors need to be corrected (for example: section 1, last paragraph “into cardiomyocytes und during β-adrenergic”, or the different font in the figure 3 legend)

We have corrected the typographical errors that have been pointed out and any additional once we have found.

  1. Table 3 and 2 appear into the text before table 1

We apologize for this mistake. We have corrected the order of appearance of the tables.

  1. It would be of value to include a figure summarizing and depicting the role of miRNAs in CVD pathophysiology along with their potential as biomarkers or therapeutic targets.

Thank you for this suggestion. We have included a summarizing figure in our manuscript. It is now figure 6 and is referred to in the discussion section.

That’s all I have only to comment for this elegant piece of work.

Thank you.

Reviewer 2 Report

In this original work Bencun et. al employ the hiPSC-CM model, to successfully computationally identify and then validate miRNA-mRNA interaction couples related to CVD pathology. The work continues the " Identification of circular RNAs with host gene-independent expression in human model systems for cardiac differentiation and disease", of the same research group.

Understanding the role of miRNAs in human disease aids the understanding of the underlying molecular mechanisms of disease development and offers novel tools for biomarker identification and therapeutic schemes development.

Therefore, the impact of this paper can be considerably high, especially currently, hence novel formulations of RNA-based medicine are being implemented through numerous vaccines. Under this lens, the miRNA sector will experience unforeseen progress in the near future.

The methodology is sound scientifically, although certain improvements should be practiced to help the reader understand, more clearly the pathway taken by the researchers to reach their conclusions (see below, a suggestion towards this aim). The discussion part is  weak and therefore I feel that it needs to be expanded, before the article is suitable for publication in the MDPI Biomedicines journal.

Hereafter my suggestions to the authors:

Minor points

In page 2 "MiRNA loci were analyzed based on miRBase v22 annotations [23]" you can avoid the capital letter for miRNA, otherwise you should also use capital at the title of 2.2.2. section.

In page 2 "We mapped small RNA-seq reads after adapter trimming and quality control to the human genome using bowtie2.3.5.1 [24]." What is quality control in this case, please expand.

In page 5, Tables 3 and 4 are not Linotype and not correctly formatted.

In page 7, in Figure 1C, the legends above need to be scaled up and the abscissa and ordinate ticks and titles.

In page 10, Figure 3 legend is not formatted in Linotype according to the journals direction for authors.

In page 12 discussion section, "Our experiments show that exogenous transfection of miRNA mimics does  " the word mimics is not clear in the syntax.

For the section discussion in page 13, the following wording is proposed to increase the impact of the beforementioned content,

as well as potentially prevent the disease itself. In addition, recent advances of mRNA product modalities manufacturing, facilitate the development of novel miRNA-based regulation treatments [44].

  1. Pharma 4.0 Continuous mRNA Drug Products Manufacturing. Pharmaceutics202113, 1371. https://doi.org/10.3390/pharmaceutics13091371

Major points

For the section Material and Methods, a general diagram is required elaborating the strategy of methods, i.e. presenting the logical order of the bioinformatic packages and the experimental validation methods, followed by the correct citation indexes.

In discussion section, the authors refer to the understanding of the molecular mechanism of cardiovascular disease (CVD) pathology. 'The identified stress-specific changes observed in the miRNA regulatory system during β-adrenergic stimulation, helps further our understanding of the molecular mechanism of cardiovascular disease (CVD) pathology'.

However, these novel findings are not then explained / justified but very briefly and not clearly, in my opinion, during the next 3 paragraphs. The authors need to expand this section accordingly.

Author Response

Reviewer #2

We are thankful for the positive and constructive feedback we have received from reviewer #2

We have revised the manuscript to address the reviewer comments. We have appended the reviewer comments with our responses highlighted in italics.

In this original work Bencun et. al employ the hiPSC-CM model, to successfully computationally identify and then validate miRNA-mRNA interaction couples related to CVD pathology. The work continues the "Identification of circular RNAs with host gene-independent expression in human model systems for cardiac differentiation and disease", of the same research group.

Understanding the role of miRNAs in human disease aids the understanding of the underlying molecular mechanisms of disease development and offers novel tools for biomarker identification and therapeutic schemes development.

Therefore, the impact of this paper can be considerably high, especially currently, hence novel formulations of RNA-based medicine are being implemented through numerous vaccines. Under this lens, the miRNA sector will experience unforeseen progress in the near future.

The methodology is sound scientifically, although certain improvements should be practiced to help the reader understand, more clearly the pathway taken by the researchers to reach their conclusions (see below, a suggestion towards this aim). The discussion part is  weak and therefore I feel that it needs to be expanded, before the article is suitable for publication in the MDPI Biomedicines journal.

Hereafter my suggestions to the authors:

Minor points

  1. In page 2 "MiRNA loci were analyzed based on miRBase v22 annotations [23]" you can avoid the capital letter for miRNA, otherwise you should also use capital at the title of 2.2.2. section.

We have changed the beginning of this sentence. It does no longer start with miRNA. (page 2 section 2.2.)

  1. In page 2 "We mapped small RNA-seq reads after adapter trimming and quality control to the human genome using bowtie2.3.5.1 [24]." What is quality control in this case, please expand.

We revised the methods section to include this description: “We mapped small RNA-seq reads after adapter trimming and quality control to the human genome using bowtie2.3.5.1 (Fig. 1) [24]. Prior to mapping, reads were filtered out from subsequent analysis if they contained uncalled bases or if they were shorter than 18bp after 3’ end quality clipping (Phred score < 10) and 3’ end adapter trimming.”

  1. In page 5, Tables 3 and 4 are not Linotype and not correctly formatted.

We have corrected the formatting.

  1. In page 7, in Figure 1C, the legends above need to be scaled up and the abscissa and ordinate ticks and titles.

We have improved the readability of figure 1C according to the reviewer’s suggestions. Figure 1C is now figure 2B.

  1. In page 10, Figure 3 legend is not formatted in Linotype according to the journals direction for authors.

We have corrected the formatting.

  1. In page 12 discussion section, "Our experiments show that exogenous transfection of miRNA mimics does  " the word mimics is not clear in the syntax.

We have changed the sentence to read: “Our experiments show that exogenous transfection of our studied miRNAs leads to a decrease in the predicted target mRNA levels in hiPSC-CMs as well as in luciferase assays.

  1. For the section discussion in page 13, the following wording is proposed to increase the impact of the beforementioned content, as well as potentially prevent the disease itself. In addition, recent advances of mRNA product modalities manufacturing, facilitate the development of novel miRNA-based regulation treatments [44].

[44] Pharma 4.0 Continuous mRNA Drug Products Manufacturing. Pharmaceutics202113, 1371. https://doi.org/10.3390/pharmaceutics13091371

As Reviewer #2 requested an expansion of the discussion section under major points, we have changed the text of this section substantially so that this suggestion. Please see below.

Major points

  1. For the section Material and Methods, a general diagram is required elaborating the strategy of methods, i.e. presenting the logical order of the bioinformatic packages and the experimental validation methods, followed by the correct citation indexes.

We have included a schematic of our workflow in the material and methods section. It is now figure 1.

  1. In discussion section, the authors refer to the understanding of the molecular mechanism of cardiovascular disease (CVD) pathology. 'The identified stress-specific changes observed in the miRNA regulatory system during β-adrenergic stimulation, helps further our understanding of the molecular mechanism of cardiovascular disease (CVD) pathology'. However, these novel findings are not then explained / justified but very briefly and not clearly, in my opinion, during the next 3 paragraphs. The authors need to expand this section accordingly.

We have revised the discussion section extensively. Please see the revised manuscript for the new version of the discussion.

Round 2

Reviewer 2 Report

The authors have addressed my remarks.